# Costs, consequences and value for money in non-medical prescribing: a scoping review

Saeideh Babashahi  ,[1] Nicola Carey  ,[2] Yogini Jani  ,[3] Kath Hart,[4] Natalia Hounsome[1]

[1]Global Health and Infection, Brighton and Sussex Medical School, University of Sussex, Brighton, UK
[2]Department of Nursing and Midwifery, University of the Highlands and Islands, Inverness, UK
[3]Centre for Medicines Optimisation Research and Education, UCLH NHS Foundation Trust and UCL School of Pharmacy, London, UK
[4]Department of Nutritional Sciences, School of Biosciences & Medicine, Faculty of Health and Medical Sciences, University of Surrey, Guildford, UK

**Correspondence to**
Dr Saeideh Babashahi;
s.babashahi@bsms.ac.uk

## ABSTRACT

**Objectives** Non-medical prescribing (NMP) is a key feature of the UK healthcare system that refers to the legal prescribing rights granted to nurses, pharmacists and other non-medical healthcare professionals who have completed an approved training programme. NMP is deemed to facilitate better patient care and timely access to medicine. The aim of this scoping review is to identify, synthesise and report the evidence on the costs, consequences and value for money of NMP provided by non-medical healthcare professionals.

**Design** Scoping review

**Data sources** MEDLINE, Cochrane Library, Scopus, PubMed, ISI Web of Science and Google Scholar were systematically searched from 1999 to 2021.

**Eligibility criteria** Peer-reviewed and grey literature written in English were included. The research was limited to original studies evaluating economic values only or both consequences and costs of NMP.

**Data extraction and synthesis** The identified studies were screened independently by two reviewers for final inclusion. The results were reported in tabular form and descriptively.

**Results** A total of 420 records were identified. Of these, nine studies evaluating and comparing NMP with patient group discussions, general practitioner-led usual care or services provided by non-prescribing colleagues were included. All studies evaluated the costs and economic values of prescribing services by non-medical prescribers, and eight assessed patient, health or clinical outcomes. Three studies showed pharmacist prescribing was superior in all outcomes and cost saving at a large scale. Others reported similar results in most health and patient outcomes across other non-medical prescribers and control groups. NMP was deemed resource intensive for both providers and other groups of non-medical prescribers (eg, nurses, physiotherapists, podiatrists).

**Conclusions** The review demonstrated the need for quality evidence from more rigorous methodological studies examining all relevant costs and consequences to show value for money in NMP and inform the commissioning of NMP for different groups of healthcare professionals.

## INTRODUCTION

Doctors have traditionally been authorised as the main group of healthcare professionals

---

## STRENGTHS AND LIMITATIONS OF THIS STUDY

⇒ This scoping review addresses an under-researched area to provide evidence on resource use and consequences (eg, service improvement, patient satisfaction, waiting times, safety, etc) of non-medical prescribing (NMP) from a large body of peer-reviewed and grey literature.

⇒ The review was limited to original studies that evaluated the economic impacts only or both costs and consequences of NMP.

⇒ Drawing conclusions on the cost-effectiveness and value for money in NMP remains difficult as the existing literature is heterogeneous with significant variation in participants, NMP types, comparators, study designs, costs and consequences evaluated.

---

to prescribe medicines.[1 2] With an increasing pace of population ageing and higher risks of chronic diseases, there is a growing demand for healthcare services and access to medicines.[3–5] Due, in part, to shortages within the medical workforce,[1 6] the authority for other healthcare professionals, such as nurses and pharmacists, to prescribe medicines has been introduced in several countries such as the USA, UK, Canada and Australia.[7–9]

Non-medical prescribing (NMP) is a key feature of the UK healthcare system that refers to the legal prescribing rights granted to nurses, pharmacists and other healthcare non-medical professionals who have completed an approved programme of education,[10–14] delivered via a variety of methods (often hybrid), including classroom teaching, one-to-one instruction, self-directed learning and e-learning.[11] NMP first emerged in the UK in 1999 for district nurses and health visitors,[2] and it came into effect for all registered nurses in 2001 and for pharmacists in 2003.[9 15 16] The UK has pioneered the gradual expansion of these prescribing roles to include a wider population of healthcare professionals across both primary care and secondary care.[8 17] Since 2005,

**BMJ**

**Table 1** Definition of main terms (and variables) used in the study

| Terminology | Definition |
|---|---|
| NMP | NMP is a term widely used in the UK, and it represents the prescribing authorities given to certain non-medical healthcare professionals (eg, nurses, dietitians, physiotherapists) after completing a prescribing training course.[9 35] |
| IP | Those using IP are responsible for assessing patients' health conditions and making decisions about patients' treatment and clinical management, including prescribing, within their scope of practice.[9 35] |
| SP | Using SP, the initial assessment and diagnosis of a patient's condition are carried out by an independent prescriber (ie, a GP or dentist), and the clinical condition is managed using a patient-specific clinical management plan agreed by the independent prescriber, supplementary prescriber and patient.[9 35] |
| PGD | PGD is a legal-written framework that allows registered healthcare professionals to supply and/or prescribe specified medicines to a predefined group of patients without them having to see a medical prescriber (eg, a GP).[7 23] |
| Medicine management or prescribing activities | A system of processes that determines how medicines are used by patients and health providers. For the purposes of this study, medicine management and prescribing activities refer to prescribing and/or the process of giving advice about medicines and the supply of medicines, as described in the research questions subsection.[17 20] |
| Cost and resource use | This refers to the direct and/or indirect medical and/or non-medical resources consumed by the study population and/or the costs associated with setting up and implementing the intervention(s) under study.[44] |
| Consequence | This refers to the health, non-health, clinical and patient outcomes representing the effects of the intervention(s) under study.[44] |
| Perspective | This refers to (one or more groups of) stakeholders' viewpoints from which economic evaluation or cost analysis is conducted.[44] Examples include the patient perspective, societal perspective or healthcare provider perspective. |
| Comparator | This refers to the alternative courses of action (eg, usual care) against which the intervention under study (eg, NMP, the subject of this study) is evaluated.[44] |

GP, general practitioner; IP, independent prescribing; NMP, non-medical prescribing; PGD, patient group direction; SP, supplementary prescribing.

physiotherapists, podiatrists, and both diagnostic and therapeutic radiographers have been able to train to become supplementary prescribers (see table 1 for definitions of supplementary prescribing (SP) and other terms).[11 18] Independent prescribing (IP) rights were subsequently granted to optometrists in 2008[14 19] and physiotherapists, podiatrists and chiropodists in 2013.[18 20 21] More recent changes in 2016 enabled therapeutic radiographers to train as independent prescribers,[22] and dietitians as supplementary prescribers[22] and in 2019 paramedics were awarded both IP and SP rights (see table 1 for a glossary of terms).

Reviews of NMP developments and its benefits in the UK and other countries have been reported by others.[10 23 24] Although NMP is embedded within UK healthcare delivery in primary and secondary care, there is still a lack of evidence regarding its value for money.[15 20 25–28] Building on an earlier review by Noblet *et al* on the effectiveness and cost-effectiveness of NMP from three randomised controlled trials (RCTs),[25] this scoping review aimed to assess a wider body of literature, including both peer-reviewed and grey literature, to identify evidence on costs and consequences and the value for money of NMP.

## METHODS

The scoping review protocol was registered with the Open Science Framework Registry on 31 July 2021 (registered DOI: 10.17605/OSF.IO/PSR3N, accessible from https://osf.io/psr3n). We followed the Preferred Reporting Items for Systematic Reviews and Meta-Analysis extension for Scoping Reviews (PRISMA-ScR) reporting guideline recommended by Tricco *et al* to report our scoping review study.[29] This scoping review was conducted using the five-stage methodological framework developed by Arksey and O'Malley and further developed by Levac *et al* and the Joanna Briggs Institute to ensure rigour in reporting the review and its methodology.[30–32] The five stages are outlined below:

### Stage 1: identifying the research questions

1. What types of prescribing practices (eg, SP, IP) have been implemented and evaluated across eligible groups of healthcare professions (eg, pharmacists, podiatrists, dietitians) in different studies?
2. What measures and tools have been used to evaluate the economic values, safety, effectiveness and other consequences of prescribing by non-medical prescribers in various settings?
3. What are relevant costs, resource use and consequences (eg, health, non-health and clinical outcomes)

associated with services provided by non-medical prescribers in both peer-reviewed and grey literature?

## Stage 2: search strategy and screening

The scope and practice of NMPs vary globally.[7 17 27 33 34] For the purposes of this review, NMP was assumed to include medicine management activities that are legally and technically considered prescribing and provided by healthcare professionals who are eligible to prescribe and have completed an approved programme of education. Consistent with Courtenay *et al* and Carey *et al*, these medicine management activities include 'making recommendations for patients to buy medicine(s) over the counter; amending prescribed medication; medication review; written recommendation to general practitioner (GP); recommending in patients' hospital notes; prescribing via hospital medication charts; patient group directions; remote prescribing via telephone, email and fax; issuing hospital-specific prescription; signing issued prescription via GP repeat prescribing system; issuing private prescription directly to the patient'.[20 27]

A comprehensive search strategy was developed by the research team to enable a stepwise search process. Based on initial exploratory research, we included grey literature and journal and conference articles with full-text written in English from 1999 to 2021.[35] On 14 January 2022, we searched PubMed, Cochrane Database of Systematic Reviews, Web of Science, Scopus and MEDLINE databases for articles published between 1 January 1999 and 1 January 2022.

The detailed search terms and strategies for different databases are presented in tables A.1 to A.5 in online supplemental data. A non-systematic search in Google Scholar was performed to find the grey literature. The search terms used for Google Scholar are equivalent to those of other search engines. In brief, our search strategy included (non-medical prescrib* OR NMP OR non-doctor prescrib*) AND (pharmac* OR nurs* OR non-medical healthcare professionals OR allied health professionals OR AHPs OR diet* OR radiograph* OR midwiv* OR physiotherap* OR podiatr* OR optometr* OR paramedic*) AND (consequences OR health outcomes OR non-health outcomes OR clinical outcomes OR effectiveness OR patient outcomes) AND (economic impacts OR costs OR resource use).

The scoping review included original research, RCT studies and grey literature analyses of resource use only or both consequences and costs to evaluate NMP provided by non-medical healthcare professionals. Commentaries, letters, protocols and editorials were excluded. A broad search strategy was implemented to ensure that the inclusion of studies was as comprehensive as possible. Search terms were derived from titles, abstracts and keywords identified in key publications and from search terms used in previous reviews related to NMP.[8 9 26] In addition, relevant references of included studies were checked (snowballing search).

All articles identified from the searches were transferred to the EndNote reference manager software V.20.2, and all duplicates and titles in languages other than English were removed. The PICOS (population, intervention, context, outcome and study design) framework was used to establish eligibility criteria.[36] Table 2 provides further information regarding the inclusion criteria according to the PICOS approach.

## Stage 3: study selection

The review process included an initial screening of the title and abstract of the studies by three authors (SB, NH and NC) to assess their eligibility for full-text retrieval. Any studies that were not excluded confidently through title and abstract screening during the initial screening step were included for full-text screening. The full-text screening of the selected studies was divided between authors and carried out independently by two reviewers (SB, NH, YJ and KH). Any disagreement on selected papers was resolved through discussion among the authors. After identifying and removing duplicates, studies were excluded if (1) they were not original studies, (2) no abstract or full-text was available, (3) they were not

| Table 2 | PICOS table describing inclusion criteria |
|---|---|
| **Component** | **Description** |
| Population | ▶ Human participants (eg, nurse, pharmacist and other non-medical prescribers, and patients with any health conditions managed by these groups)<br>▶ No restriction on age or gender |
| Intervention | All types of non-medical prescribing (any medicine management activity that is legally and technically considered prescribing and provided by non-medical professionals) |
| Context | ▶ All peer-reviewed published articles (in journals and conferences) and grey literature with full-text written in English from 1999 to 2021<br>▶ No restrictions on setting or country |
| Outcome | Cost and consequence outcomes of NMP services provided by nurses, pharmacists and other non-medical prescribers |
| Study design | Original research and clinical trials that evaluated costs and economic impacts of NMP only or both cost and consequence outcomes of NMP (peer-reviewed or grey literature) |

NMP, non-medical prescribing.

in English, or (4) the focus of the study was outside the scope of our review (see table 2), or (5) prescribing and medicine management activities evaluated did not meet those indicated by Courtenay *et al* and Carey *et al*.[20 27]

### Stage 4: data extraction and analysis

Data from the articles and grey literature based on the inclusion criteria mentioned above were extracted using a bespoke data extraction form. A Microsoft Excel 2019 based form was initially developed by the first author and validated by other authors for charting the data from selected studies and reporting the variables regarding the study, participants, interventions and outcome characteristics—for example, authors, publication year, study context and design, sample size, type of prescribing, cost and consequence outcomes measures and key findings—based on our research questions (table 1 represents the definition of the main variables). Data extracted were checked by a second reviewer for accuracy and completeness.

### Stage 5: collating and reporting the results

The PRISMA-ScR reporting checklist (table B, online supplemental file 1) was used to synthesise and report the results of our scoping review.[29] Data synthesis was undertaken by the first author in consultation with the research team. The findings of selected studies were summarised and presented in tabular forms and descriptively highlighting the key research findings (eg, economic impacts, consequences of NMP, setting, NMP type) of selected studies and the existing research gaps around NMP practice.

### Patient and public involvement

No patients or public were involved in the study.

## RESULTS

### Database search findings

The database search generated 420 records. A total of 236 records were removed due to duplication. Of the remaining 184 records, we excluded 171 records in the initial review of titles and abstracts as these studies were not original research evaluating NMP. For the remaining 13 records, the full-text papers were independently reviewed by 2 reviewers, and a further 4 studies were excluded because they did not report resource use and economic impacts of NMP or did not fit within our definition of prescribing and medicine management activities. Nine studies were included in the final review (eight original research studies[7 20 34 37–41] and one grey literature paper).[42] Figure 1 shows the PRISMA flow chart of the included studies in our scoping review.

### General characteristics of included studies

The key characteristics of included papers are summarised in table 3 and table C (online supplemental data). Of the nine papers, six were from the UK,[7 20 34 40–42] two from Canada[37 39] and one from Australia.[38] Papers were

published between 2010 and 2022 and evaluated the impact of NMP practices by pharmacists (n=4),[37–40] nurses (n=3),[7 34 41] physiotherapists and podiatrists (n=1),[20] and another estimating NMP cost-savings in primary and secondary care for a range of health professions.[42] Types of prescribing services evaluated in these studies included SP (n=2)[41 42] or IP (n=8)[7 20 34 37–40 42] and community nursing.[42]

### Methodological and reporting considerations

Three out of nine papers conducted a model-based economic evaluation (ie, cost-effectiveness analysis) using the outcomes from an earlier trial with an assessment of uncertainty (in the form of a deterministic and/or probabilistic sensitivity analysis).[37–39] Four studies conducted a cost–consequence-based approach listing costs and outcomes of NMP without assessing sources of uncertainty.[7 20 34 41] A bottom-up costing approach (using detailed input data from records or questionnaires at the service provider level) was used in most studies with clear information on costs per unit.[7 20 34 37–40] Overall non-model-based studies did not provide an explanation of sample size sufficiency. Only one study suggested that determining an optimal and larger sample size would be required to draw a precise and accurate conclusion.[40] Two studies failed to specify the number or characteristics of the study participants (eg, non-medical prescribers or patients).[40 41]

### Measures of costs of NMP

The resource use and costs evaluated in the included studies fall into the following three main categories:

#### Prescribing training course

Four out of nine articles applied direct costs associated with prescribing training and NMP courses (eg, training course fee, supervision time, employer-paid study time),[7 37 39 40] with one study using time-off-work to complete the course.[41] Other relevant expenses such as out-of-pocket expenses (eg, travel, accommodation) by qualified non-medical prescribers and their (unpaid) personal study time were included in two studies.[7 41]

#### Prescription (and consultation)

Expenses applied in this category by some studies included tests and other relevant services, referrals to other healthcare professionals, frequency of follow-up, time spent preparing for a prescribing consultation, time taken to prescribe, review or complete the medication plan for a patient, and the number of patients prescribed for, consultation frequency, time spent discussing the patient and obtaining prescriptions or clinical advice sought from GPs or other NMP practitioners, unplanned consultations for the health condition after the index consultation and frequency of new medications.[7 20 34 37 38 40 41] Incorrect or overprescribing was identified and considered as an indication of wastage, and the 'wasted' medication, as well as underprescribed medicine that should

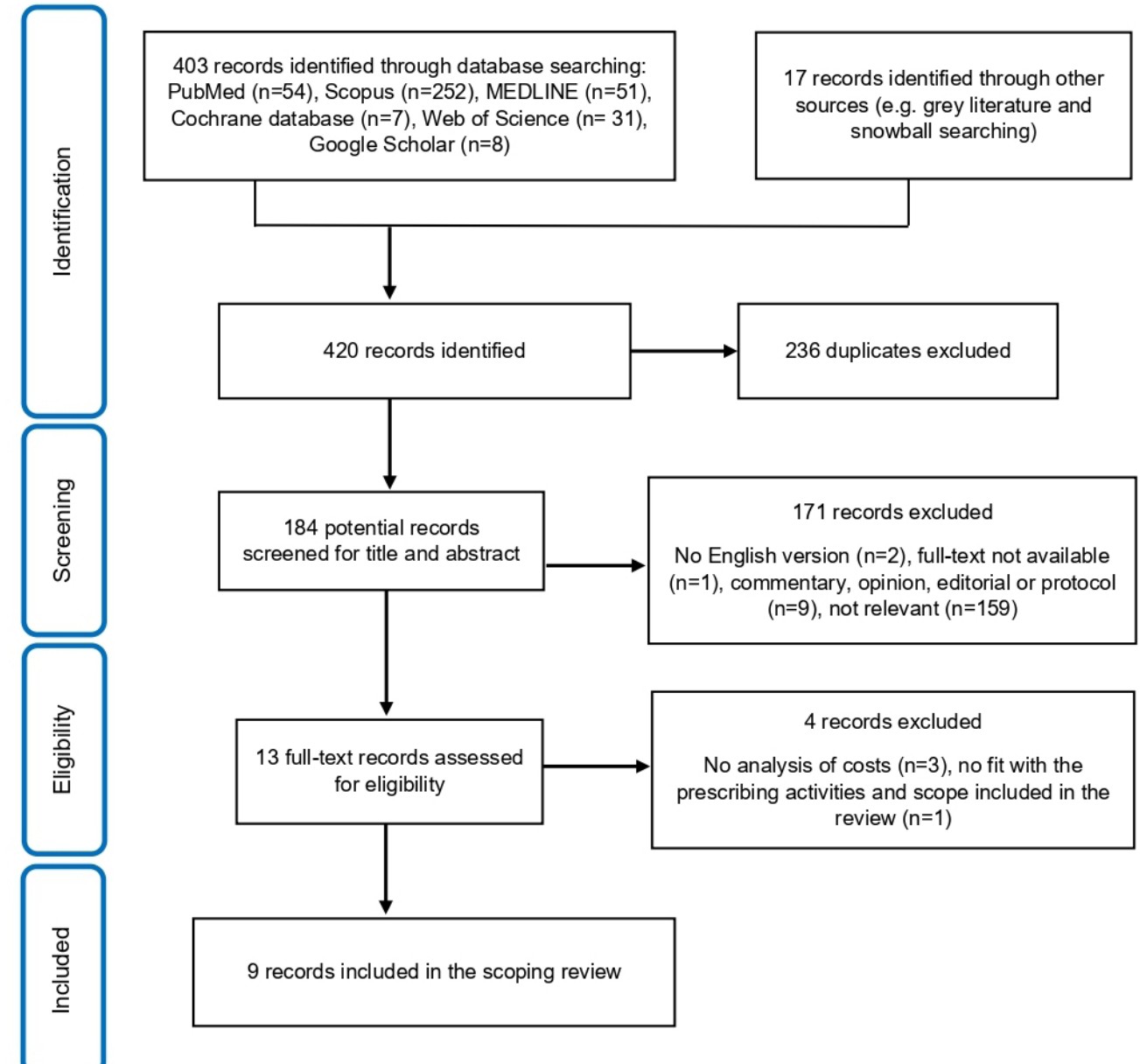

**Figure 1** PRISMA flow chart of the study selection process. PRISMA, Preferred Reporting Items for Systematic Reviews and Meta-Analyses.

have been prescribed, were considered as another source of cost.[7 41]

### Other relevant expenses
Some studies also considered the expenses associated with service utilisation, for example, hospital admissions, outpatient expenses, inpatient days and A&E visits,[20 34 39–41] or the healthcare and medical costs associated with targeted health conditions across case and control groups.[37–39]

### Measures of consequences of NMP
Health-related quality of life was one of the main health outcomes evaluated using EQ-5D (European Quality of Life Five Dimensions) or SF-6D (Short Form Six Dimensions)

questionnaires,[20 37–40] and the benefits to patients of appropriate prescribing were measured in terms of increased quality-adjusted life-years (QALYs) in some of these studies.[37–40] Multiple studies evaluated patient experience and satisfaction as one of the main patient outcomes.[7 20 34 41] Life years gained were applied by two studies.[37 39] Medicine adherence and ease of access to services were other outcomes reported in one study.[20] Examples of specific clinical and health outcomes used in the studies were self-care and relevant clinical indicators such as Hemoglobin A1c test results (mean blood sugar level) and body mass index for patients with diabetes,[34] the reduced risk of the disease under study (eg, venous thromboembolism, cardiovascular disease (CVD))[37–39] and reduced blood pressure.[39]

**Table 3** Overview and general characteristics of included studies

| Authors (year) | Context (country/setting) | Study design | Type of prescribing | Comparator | Medicine management or prescribing activity | Study population |
|---|---|---|---|---|---|---|
| Black et al (2022)[7] | UK, urban sexual health services | Mixed-methods and a comparative case study (cost-consequence framework) | Nurse IP | PGD by non-prescribing nurses | Prescribed medications | N=26 nurse prescribers N=67 PGDs users |
| Carey et al (2020)[20] | UK, mixed range of settings (primary and secondary care, social enterprise and private practice) | Mixed-methods and a comparative case study (cost-consequence framework) | Physiotherapist IP Podiatrist IP | Non-prescribing physiotherapists Non-prescribing podiatrists | Prescribed and reviewed medications | N=488 patients (243 IP sites and 245 NP sites) N=7 matched pairs of IP and NP sites (3 podiatrists and 4 physiotherapists) |
| Al Hamarneh et al (2019)[37] | Canada, primary care (cardiovascular risk reduction) | Cost-effectiveness analysis (Markov model) | Pharmacist IP | Usual care | Prescribed and reviewed medications | The authors developed their model based on the population observed in the $R_x$EACH trial as follows: N=723 patients (370 in intervention and 353 in control) N=54 pharmacies in the RCT[45] |
| Hale et al (2018)[38] | Australia, an elective surgery preadmission clinic (venous thromboembolism) | Cost-effectiveness analysis (decision tree model) | Pharmacist IP | Usual care | Prescribed medications | The authors developed their model based on the population observed in an earlier trial as follows: N=384 patients (194 in intervention and 190 in control) N=1 pharmacist prescriber N=59 medical prescribers[46] |
| Marra et al (2017)[39] | Canada, community care, hospitals or primary care (hypertension) | Cost-effectiveness analysis (Markov model) | Pharmacist IP | Usual care | Prescribed medications | The authors developed their model based on the population observed in the $R_x$ACRION trial as follows: N=248 patients (181 in intervention and 67 in control) N=20 pharmacists practised in the community N=2 pharmacists from hospital outpatient clinics N=6 pharmacists from primary care clinics[47] |
| i5 Health (2015)[42] | England, various settings (eg, primary and secondary care) | Economic analysis of audits, self-reported questionnaires, interviews | IP and SP (for a range of professions, for example, physiotherapists, podiatrists, midwives and radiographers) Community nurse prescribers | NA | NA | Based on an estimation of the NMP practitioners registered with Northwest England NHS trusts (N=1566 unique prescribers) |

**Table 3** Continued

| Authors (year) | Context (country/setting) | Study design | Type of prescribing | Comparator | Medicine management or prescribing activity | Study population |
|---|---|---|---|---|---|---|
| Courtenay et al (2015)[34] | England, primary care (type 2 diabetes) | Mixed-methods and a comparative case study (cost-consequence framework) | Nurse IP | Non-prescribing nurses | Prescribed and reviewed medications, recommended decisions, provided advice and discussed medications with GPs or colleagues | N=12 general practices (6 prescribing nurses and 6 non-prescribing nurses) N=214 patients (131 in nurse prescriber sites and 83 in non-prescriber sites) |
| Neilson et al (2015)[40] | UK, primary care (chronic pain) | Regression analysis of costs and effects; the expected value of sample information analysis | Pharmacist IP | Usual care | Prescribed and reviewed medications | N=6 general practices N=125 patients (39 in prescribing, 44 in review and 42 in usual care arms) No information is provided about the number of non-medical prescribers in the two groups |
| Norman et al (2010)[41] | UK, primary care (mental health) | Cost-consequences analysis; matched post-test control study | Nurse SP | Usual care | Prescribed medicines | N=90 patients (45 matched pairs) No information is provided about the number of prescribers in the two groups |

GP, general practitioner; IP, independent prescribing; NA, not available; NHS, National Health Service; NMP, non-medical prescribing; NP, non-prescribing; PGD, patient group direction; RCT, randomised controlled trial; SP, supplementary prescribing.

## Key findings: the costs and consequences of NMP

A summary of the key cost and consequence findings is provided for pharmacists, nurses and other non-medical prescribers in turn (see table C in online supplemental data for detailed information).

## Pharmacists

The NMP practices by pharmacists were evaluated across a range of health conditions (eg, venous thromboembolism, hypertension), and significant improvements in health and clinical outcomes were reported at the end of the observation in three studies.[37–39] As such, Marra *et al* found the 30-year risk of CVD in the pharmacist prescriber group was reduced from 0.61 in base case to 0.41 (indicating a reduction of two CVD events in every 10 individuals receiving the intervention).[39] Although the intervention was associated with increased costs of C\$7145 due to the intervention itself and medications, this was compensated for by a reduction of C\$15094 in CVD and other comorbidities costs, suggesting pharmacist IP was less costly and more effective than usual care.[39]

Consistent with Marra *et al*, two other studies—that is, Al Hamarneh *et al* and Hale *et al*—reported that pharmacist prescribing was cost-effective and cost saving for patients with CVD and venous thromboembolism, respectively.[37 38] Only Neilson *et al* found that, relative to the usual-care arm, pharmacist prescribing for chronic pain was more costly (£77.5 for prescribing and £54.4 for review arms) and provided similar QALYs. Neilson *et al* recruited a total sample of 125 patients in this RCT, but the authors recommended a larger sample size (between 460 and 690 for a threshold of £30000 QALY gained or 540–780 for a threshold of £20000 QALY gained) according to an expected value of sample information analysis (indicating that additional information collected from a larger sample will reduce uncertainty and provide more reliable data).[40]

## Nurses

Norman *et al* indicated that patients in the mental health nurse prescriber group had a significantly higher level of satisfaction with nurse prescribers than those in the medical prescriber group.[41] Similarly, Courtenay *et al* reported that the average patient satisfaction for some specified aspects of care was significantly higher among diabetic patients in the nurse prescriber group than among those of the non-prescribing nurses.[34] Nonetheless, no significant differences were reported with respect to patients' overall satisfaction by Courtenay *et al* and Black *et al*. Other specific or generic health and social outcomes were found to be similar among nurse prescribers and the control groups in these three studies.[7 34 41]

NMP was deemed resource-intensive for both providers and nurse prescribers. According to Black *et al*, the training-related costs included the course fee (paid fully by employers or training grants—ranging from £900 to £3555 in 2016), an average of 20.1 of employer-paid study days for 92% of nurse prescribers (ranging from 1

to 31 funded by employer), and an average of 7.4 supervised days (ranging from 2 to 13.7 day, incurring a cost of £6451 to the NHS (National Health Service) for each nurse—ranging from £1283 to £11 138) during training.[7] It is important to note that although PGDs (ie, patient group directions, please see table 1 for more information) provide a legal framework for health professionals to supply and administer a specified medicine to a predefined group of patients, and there is no mandatory training required prior to their use; there are limitations to their use, indicating that NMP might be worth the training cost.[7 23] Employment costs of prescribing nurses were deemed potentially higher as they were on higher pay bands compared with non-prescribing nurses, including PGD users.[7 34 41] Consultation durations and unplanned reconsultations were similar for both sexual health nurse prescribers and medical prescribers, as reported by Black *et al*.[7] However, Courtenay *et al* reported longer consultations for patients with diabetes managed by nurse prescribers suggesting it was more costly relative to GPs and non-prescribing nurses.[34] No statistically significant differences in prescribing new medicines or use of other healthcare services between groups were identified in any of the three studies assessing this outcome.[7 34 41]

## Other non-medical prescribers

Only one study evaluated the benefits and costs of services by physiotherapist and podiatrist independent prescribers compared with non-prescribing physiotherapists and podiatrists.[20] Carey *et al* showed the level of satisfaction with consultation and services was significantly higher in both non-medical prescriber groups. Patients of physiotherapist or podiatrist independent prescribers were more likely to receive medicine information or advice during consultations (39.7%) compared with patients managed by non-prescribers (24.5%). No significant differences were reported in the quality of life in patients for all groups.[20] Consultation durations were longer for both prescriber groups, resulting in increased costs for prescribing physiotherapists (£7.95 per contact) and prescribing podiatrists (£8.62) compared with non-prescribers. No training-related costs were reported.[20]

## DISCUSSION

Building on a previous systematic review[25] which included only three RCTs published before 2015, we have included a wider range of studies evaluating the consequences, resource use, costs and value for money in NMP. We used the PRISMA-ScR framework to guide the review, searched multiple databases and used snowballing techniques to improve the comprehensiveness of the study. Despite this, only one additional source of evidence from the grey literature was identified. The NMP literature has largely focused on assessing the benefits and effectiveness of prescribing authorities without evaluating the costs and resource use. Some other studies have concentrated on topics such as NMP trends and related national

policies over time or implementation barriers and/or facilitators of NMP for different professions.[1 9 17 24 43] This review demonstrated the lack of evidence on costs, consequences and value for money in NMP by different groups of healthcare professionals.

Our scoping review identified nine sources of evidence that evaluated the economic impacts, resource use and consequences of NMP. Three studies showed pharmacist prescribing was superior in all outcomes and cost saving at a large scale.[37–39] Others reported similar results in most health, clinical and patient outcomes across other non-medical prescribers and control groups (eg, GP-led usual care). NMP was deemed resource intensive for both providers and other groups of non-medical prescribers (eg, nurses, physiotherapists, podiatrists).

This scoping review revealed evidence sources were heterogeneous with regard to design, setting, range of cost and consequence outcomes, NMP types and comparators. In general, the existing evidence indicates that services provided by non-medical prescribers might positively influence patients' satisfaction with care, medication and their quality of life.[20 34 37–39] However, some of these findings came from non-RCT studies without robust evaluation of all relevant consequences and costs. In addition, some of these studies recruited small sample sizes, suggesting it is difficult to make any statement about the significance of the results beyond the sample included, and therefore, these findings should be treated with caution.

The costs and consequences evidence on NMP has slowly grown since 2010 and appears to be concentrated in three countries, the UK, Canada and Australia, where prescribing rights are more developed. Despite the large increase in NMP in the UK and around the world and the increasing number of studies on NMP, there is still very limited information on the effectiveness, costs and cost-effectiveness of NMP by different professions. Many papers evaluated NMP delivered by nurses and pharmacists using various sources of costs, health, clinical and patient outcomes with varied comparators for a range of health conditions, which limits their generalisability and usefulness for other settings and professions. Only three studies conducted a cost-effectiveness analysis to evaluate and demonstrate value for money in pharmacist prescribing.[9 37 39] Other studies used a cost–consequence approach (CCA) that provided disaggregated costs and outcomes of NMP for nurses, physiotherapists and podiatrists.[7 20 34] Although CCA helps identify and list relevant costs and outcomes associated with the interventions, it does not provide a definitive cost-outcome ratio and definite cost-effectiveness results for the interventions under study.[44]

The number of studies, particularly economic evaluation studies, assessing the economic burden and effectiveness of NMP has been increasing, but there is still a dearth of evidence on the cost-effectiveness and value for money in NMP authorities by recently awarded non-medical prescribers such as radiographers and dietitians.

As most cost-effectiveness evidence relates to pharmacists, it is important to evaluate the impact, safety, resource use and economic value of prescribing by non-medical prescribers in other professions to inform policy and practice around NMP where it provides value for money. It is also important to acknowledge and further explore the challenges related to capturing these data, as NMP has been introduced as an additional role for healthcare professions, and hence it is not easy to separate and capture some of the added costs and values in terms of these additional prescriptive authorities.[20 25]

There seem to be research-quality gaps in the literature. Although we did not assess the quality of included studies, some of the studies have performed non-model-based analysis using small samples that might affect the analysis, and in some cases, the main outcomes and sources of costs (eg, training related) were not included in the analysis. Despite the importance of rigour in quantitative research, sample size reporting and sufficiency assessment remained inconsistent and partial in these studies.

## STRENGTHS AND LIMITATIONS

A rigorous search was conducted, allowing for a diverse set of literature (from both peer-reviewed and grey) to be identified in a robust and reproducible manner. To our knowledge, this is the first scoping review covering and representing the largest and most up-to-date evidence on the costs and consequences of prescribing practices by nurses, pharmacists and other non-medical prescribers. This scoping review contributes to the discussion of the costs and consequences of NMP and the existing research gaps regarding value for money in NMP for different groups of healthcare professionals. Original studies that did not report resource use and costs associated with NMP were not included in our review. While this strategy contributed to a more focused search, studies that reported only the effectiveness and benefits of NMP practices without evaluating costs are missing. Comparison of studies was challenged by heterogeneity regarding the profession, type of NMP, costs and consequences evaluated.

## CONCLUSION

NMP practice is now an integrated feature of healthcare delivery in the UK and around the world, but considerable uncertainty remains regarding the costs, consequences and cost-effectiveness of the prescribing rights granted to non-medical prescribers, including therapeutic radiographers and dietitians. In order to determine accurate mean values and detect cost and benefit differences across non-medical prescribers and control groups, it is important that future studies involve larger and more representative samples with greater power. Adopting a model-based approach within each profession using targeted outcome measures would also enable a more robust comparison and improve understanding of how to best use NMP and

healthcare professionals' skills and ensure it offers a cost-effective solution to providing faster and improved access to medicine and healthcare services for patients by the most appropriate healthcare professionals.

**Contributors** SB: study design; formulation of search strategies; investigation, screening and review—titles, abstracts and full texts; data extraction; writing—original draft, review and editing; guarantor; NC, YJ, KH and NH: Study design; formulation of search strategies; screening and review—full texts, data extraction; writing—review and editing. All authors read and approved the final version of the manuscript before submission.

**Funding** This scoping review is conducted as part of the health economic evaluation in the TRaDiP project that was funded by the Department of Health Policy Research Programme, the National Institute for Health Research (grant number: PRP12697/86202356).

**Competing interests** None declared.

**Patient and public involvement** Patients and/or the public were not involved in the design, or conduct, or reporting, or dissemination plans of this research.

**Patient consent for publication** Not applicable.

**Provenance and peer review** Not commissioned; externally peer reviewed.

**Data availability statement** All data are available within the article and online supplemental materials.

**ORCID iDs**
Saeideh Babashahi http://orcid.org/0000-0003-2204-2028
Nicola Carey http://orcid.org/0000-0003-2841-1760
Yogini Jani http://orcid.org/0000-0001-5927-5429

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
