## [Reviewer comments · BMJ Open]

ARTICLE DETAILS

TITLE (PROVISIONAL)	Costs, consequences and value for money in non-medical prescribing: A scoping review
AUTHORS	Babashahi, Saeideh; Carey, Nicola; Jani, Yogini; Hart, Kath; Hounsome, Natalia

VERSION 1 – REVIEW

REVIEWER	McIntosh, Trudi Robert Gordon University, Pharmacy and Life Sciences
REVIEW RETURNED	20-Sep-2022

GENERAL COMMENTS	Costs, consequences and value for money in non-medical prescribing: A scoping review Thank you for preparing and submitting this scoping review which I have read with great interest. It addresses an important aspect of non-medical prescribing and provides very useful information, addressing a gap in current knowledge. It is very well written with only a few minor errors or omissions in the main text. There appear to be several errors in citing and listing references. Please check all of these very carefully against the journal's requirements. Please be consistent in use of terminology: there appears to be a difference in the aim as stated in the title and abstract, and in the main text. Title and abstract aim: Costs, consequences and value for money in non-medical prescribing Aim within text: resource use, cost and consequence outcomes of non-medical prescribing. The authors' guidance says Authors are encouraged to submit figures and images in colour - there are no colour charges. We require that you upload your figures as separate files rather than embedding them in the manuscript. I would also encourage use of appropriate colours in your tables. Figure1 appears to be part of the manuscript but it may be that this has been done to aid reviewers. I have attached my comments on your manuscript . I can't see the comments on the version I've uploaded and hope that you can see them. Editor: please let me know if my comments aren't visible and I'll try again. Thanks.
---

REVIEWER	Graham-Clarke, Emma Sandwell and West Birmingham Hospitals NHS Trust, Anaesthetic Department
REVIEW RETURNED	22-Sep-2022 Thank you for inviting me to review this paper. As the

	authors, and Noblet et al, report there is a dearth of evidence in this important field. I have a few comments to make regarding the paper. I note that you registered the protocol in advance, and published it, as well as following the PRISMA-ScR guidelines and I commend you for that. The authors comment correctly that optometrists are non-medical prescribers, but appear to group them under the more general 'allied health professional' (AHP) heading. They are not classed as AHPs (NHS England - https://www.england.nhs.uk/ahp/role/), and they aren't regulated by the HCPC, which regulates AHPs, but are in fact regulated by GOC. Please ensure therefore that you refer to optometrists separately throughout the paper. If you could find no studies relating to optometrists, then it would be valuable to state that. It was difficult to relate the results section to your very clear research questions that you set out initially. Please rearrange the results section to reflect the order of the questions, as that would aid the reader. Please check your use of abbreviations and ensure that they are defined the first time they appear in the text (eg CVD and HbA1C on page 14) Please check that any references to supplementary data state that, e.g. ref to Table B - Page 9, line 42 and Table C - Page 15, line 9. Page 4, lines 20-27. Your reference numbers jump from 9 to 27. Please check you references as in the numbered system they should follow sequentially when first referenced. Please also ensure that every mention of a study is referenced (e.g. page 15, lines 41-45) Please check that any references to supplementary data state that e.g. ref to Table B - Page 9, line 42 and Table C - Page 15, line 9. Page 4, Line 44 – 'optometrists' repeated twice Page 4, Line 46 – comma after '2016' not required Page 6, Line 54, Stage 2. You comment that the scope of NMP ranges and say 'ranging from a restricted formulary to... please clarify what you mean by that. Page 7, lines 3-14. Issuing of new prescriptions via a GP system is specifically not mentioned – are these activities not included? (the only GP system related activity appears to be repeat prescribing). Page 9, line 3 – you comment that two reviewers screened the full text articles, and then list 4 reviewers. Please clarify – for example were they split between the authors. Page 9, lines 46-52. As it currently reads, it implies that a separate descriptive report was written, in addition to the table. If you mean by 'descriptive report' the narrative in the table, then it would read better as '...tabular form, with a descriptive report...' Page 11-12, Table 3. The legend includes several abbreviations that
--	---

	are not used in the table (but are used in Table C in the supplementary data). Please make sure the legends match the relevant table. Page 13, line 5. If table C is part of the supplementary data, then this line isn't required as it will be cross linked from the previous reference. Page 16, second paragraph. Your comments about PDs not requiring specific training could be interpreted that PDGs are better as the costs are less. Is this correct, or are there limitations to PDGs that mean that NMP is worth the training cost? Page 16 line 37 – comma after PDGs not required Page 19, lines 10-16. I'm not clear that your results show that NMP provides value for money (particularly in the nursing group, which comprise the greatest number of NMPs). Please clarify what you mean by this sentence. Page 19, line 44-49. The studies you include form before 2015 were not RCTs, and hence were excluded from the paper by Noblet et al. It would be more appropriate to state that you have included '...findings from studies that were published...' and '... as not RCT studies, were excluded from the previous review...' or something similar.
--	--

VERSION 1 – AUTHOR RESPONSE

Reviewer #1

Comment #1: Costs, consequences and value for money in non-medical prescribing: A scoping review.

Thank you for preparing and submitting this scoping review which I have read with great interest. It addresses an important aspect of non-medical prescribing and provides very useful information, addressing a gap in current knowledge. It is very well written with only a few minor errors or omissions in the main text. There appear to be several errors in citing and listing references. Please check all of these very carefully against the journal's requirements.

Authors' response: Thank you for your remarks and comments. Following your suggestions, we have been able to improve the paper. Below, we explain how we have dealt with your comments. The changes are all highlighted in yellow here and in the marked version of the manuscript and supplementary data.

Comment #2: Please be consistent in use of terminology: there appears to be a difference in the aim as stated in the title and abstract, and in the main text.

Title and abstract aim: Costs, consequences and value for money in non-medical prescribing
 Aim within text: resource use, cost and consequence outcomes of non-medical prescribing.

Authors' response: Thanks for your comment. This was revised where relevant according to your comment. For example, see the marked manuscript, page 5, last line:

"... this scoping review aimed ... to identify evidence on costs and consequences and the value for money of non-medical prescribing."

Comment #3: The authors' guidance says: Authors are encouraged to submit figures and images in colour - there are no colour charges. We require that you upload your figures as separate files rather than embedding them in the manuscript.

I would also encourage use of appropriate colours in your tables. Figure 1 appears to be part of the manuscript but it may be that this has been done to aid reviewers.

Authors' response: Thanks for your suggestion. Figure 1 (partly in colour) was not embedded in the manuscript, and it was uploaded as a separate pdf file in line with the journal submission process.

Comment #4: I have attached my comments on your manuscript. I can't see the comments on the version I've uploaded and hope that you can see them.

Editor: please let me know if my comments aren't visible and I'll try again. Thanks.

Please see attached report

Authors' response: Thank you. We have listed your other comments (and our responses) below.

Comment #5: Is it correct to have citations after the full stop? Please check and amend if necessary. I won't comment again. (Page 4, Introduction, 1st paragraph)

Authors' response: This is in line with the journal submission guideline: "Reference numbers in the text should be inserted immediately after punctuation (with no word spacing)."

Comment #6: Citations out of sequence. Please amend and check all. Another citation out of sequence. I won't comment again but please check every one carefully. (Page 4, Introduction, 2nd paragraph)

Authors' response: All citations/references were checked, and they are all in order now.

Comment #7: Did it come into effect then? 2003? Please check.

Authors' response: Thanks for your comment. It was changed to 2003 (marked manuscript, page 4, paragraph 4, line 15).

Comment #8: Please explain this term and independent prescribing by NMPs - your readers may not be familiar with it. (Page 4, Introduction, 2nd paragraph, line 42)

Authors' response: The definitions of the main terms (including independent prescribing) are provided in Table 1, next to this text (page 4, paragraph 2, lines 18 and 19). So we have referred the readers to Table 1 as follows:

"... (see Table 1 for definitions of supplementary prescribing (SP) and other terms)"

Comment #9: I hope this table will be printed adjacent to this text - lots of important information.

Authors' response: We hope the Journal places 'Table 1' next to this text.

Comment #10: Certain (Page 5, Table 1, NMP row, Line 7)

Authors' response: This (i.e. certain) was added to the sentence as follows (Page 5, Table 1, 'NMP' row):

"... and it represents the prescribing authorities given to certain non-medical healthcare professionals ..."

Comment #11: stakeholder's. One stakeholder's viewpoint, two stakeholders' viewpoints. (Table 1, Stakeholder, Line 33)

Authors' response: This was revised as follows (Page 5, Table 1, 'Perspective' row):

"This refers to (one or more groups of) stakeholders' viewpoints from which economic evaluation or cost analysis is conducted.26 Examples include the patient perspective, societal perspective or healthcare provider perspective."

Comment #12: Why use square brackets? you've also used more usual ones. Is there a reason for using two types? (Page 5, Line 49)

Authors' response: This was revised, and we have used 'parentheses' throughout the manuscript.

Comment #13: Meaning? Seems an odd term. If there isn't a clear reason for using it I suggest you find something else., (Page 5, Line 53)

Authors' response: The 'state-of-the-art' was removed from this sentence.

Comment #14: In your abstract you say:

The aim of this scoping review is to identify, synthesise and report the evidence on the costs, consequences and value for money of NMP provided by non-medical healthcare professionals..

Please ensure this all matches. (Page 5, Line 56)

Authors' response: This was revised here and in other places, as already explained in our response to Comment #2.

Comment #15: I'm not clear what the difference is here. Presumably someone could write a prescription from a restricted formulary? (Page 6, Line 54)

Authors' response: This was removed to prevent confusion.

Comment #16: consequences? wold match with title. Meaning of 'effects' is not clear to me. (Page 5, Line 55)

Authors' response: The term 'consequences' was used instead (manuscript, page 7, last paragraph).

Comment #17: languages other than English? (Page 8, Line 13)

Authors' response: This was revised as follows (manuscript, page 8, paragraph 2):

"... all duplicates and titles in languages other than English were removed."

Comment #18: I think I know what you mean, but you could have a NMP service that was managed by a senior nurse, pharmacist or AHP but not provided by them. Do you mean 'provided by'? (Page 8, Table 2, Outcome row, Line 38)

Authors' response: This was revised, and 'provided' was used instead.

Comment #19: were. Data is a plural (Page 9, Line 20)

Authors' response: This was revised and 'were' was used instead.

Comment #20: Need superscript (Page 9, Line 22)

Authors' response: The sign '©' was removed.

Comment #21: Citation missing? (Page 10, Line 50)

Authors' response: Citation was added.

Comment #22: decision tree - no need for capital letter (Table 3, Page 11, Line 22)

Authors' response: This was revised to 'decision tree'.

Comment #23: Please write in full (Table 3, Page 12, Line 21)

Authors' response: The full term (i.e. expected value of sample information analysis) was added.

Comment #24: Sorry, I don't know what this is. Please would you explain (Page 13, Line 22)

Authors' response: This was clarified as follows:

"A bottom-up costing approach (using detailed input data from records or questionnaires at the service provider level) was used ..."

Comment #25: Is this those in training or once qualified? (Page 13, Line 54)

Authors' response: This was revised, and 'qualified' was added.

Comment #26: Is this different from 'study time' above? (Page 13, Line 54)

Authors' response: This was revised. The study time mentioned in the above line is 'fully-funded (provided) by the employer', while there might be personal study time that is not funded by the employer, and NMP-trained professionals have to use their own personal time (so the 'personal' study time is considered among the expenses in the out-of-pocket expenses category).

Comment #27: Meaning not clear (Page 14, Line 3)

Authors' response: This was clarified as follows:

"Expenses applied in this category by some studies included tests and other relevant services, referrals ..."

Comment #28: Tautology. AHPs is sufficient (Page 15, Line 9)

Authors' response: This was revised, and 'AHPs' was used instead.

Comment #29: Please explain this more fully (Page 15, Line 52)

Authors' response: This was clarified as follows:

"... expected value of sample information analysis (indicating that additional information collected from a larger sample will reduce uncertainty and provide more reliable data)."

Comment #30: With what? (Page 16, Line 8)

Authors' response: This was clarified as follows:

"... that patients in the mental health nurse prescriber group had a significantly higher level of satisfaction with nurse prescribers than those ..."

Comment #31: Sorry, I don't understand the difference between this and the previously stated number of paid study days. (Page 16, Line 33)

Authors' response: This was clarified as follows:

"... an average of 20.1 employer-paid study days were reported by 92% of nurses and an average of 7.4 supervised days for each nurse (an average cost of £6,45 to the NHS per nurse) during training."

Comment #32: Please re-write this sentence to remind the reader what PGDs are and how they're used before making your point. (Page 16, Line 35)

Authors' response: This was revised as follows:

"... PGDs (i.e. patient group directions, please see Table 1 for more information) provide a legal framework for health professionals to supply and administer a specified medicine to a pre-defined group of patients ..."

Comment #33: First mention? (Page 17, Line 39)

Authors' response: A line about this (i.e. snowballing search) was added to the method section (manuscript, page 8, paragraph 1) as follows:

"In addition, relevant references of included studies were checked (snowballing search)."

Comment #34: Please re-write this sentence - some duplication. (Page 17, Line 54)

Authors' response: This was revised, and the duplication was removed from this sentence.

Comment #35: Not clear re 'control groups'. Please clarify what you mean. (Page 18, Line 5)

Authors' response: An example (e.g. GP-led usual care) was provided to clarify this.

Comment #36: I don't think you include pharmacists in this but please make clearer. (Page 18, Line 10)

Authors' response: This sentence does not include pharmacists. The NMP evidence for pharmacists indicated that NMP was cost-saving for this group. But NMP was found resource-intensive for other groups of non-medical prescribers (e.g. nurses, physiotherapists, podiatrists).

Comment #37: On what? (Page 18, Line 39)

Authors' response: This was added as follows:

"... there is still very limited information on the effectiveness, costs and cost-effectiveness of NMP by different professions."

Comment #38: Please re-write - not clear (Page 19, Line 7)

Authors' response: This was revised and clarified as follows:

"... but there is still a dearth of evidence on the cost-effectiveness and value for money in NMP authorities by recently awarded non-medical prescribers such as radiographers and dietitians."

Comment #39: were not? Or not clear what this means. (Page 19, Lines 45-47)

Authors' response: The lines here (about Noblet et al) were removed.

Comment #40: Is faster access the only reason for promoting NMP? What about improving access to care (not just faster access), ensure the patient is seen by the most appropriator person. making best use of HCP's skills? (Page 20, Lines 32-34)

Authors' response: The sentence was revised according to your comment as follows:

"... improve understanding how to best utilise NMP and healthcare professional's skills and ensure it offers a cost-effective solution to providing faster and improved access to medicine and healthcare services for patients by the most appropriate individuals."

Comment #41: Please ensure these all match the journal requirements. There are page numbers missing and there appears to be a mixture of abbreviated and full journal titles. Please check all. Please also check journal requirements for referencing resources accessed online e.g. Ref no.13 - publisher details are missing here and in several other references.

Authors' response: This was checked here and in other places and the references were updated where relevant.

Comment #42: Ref 4: Page number missing?

Authors' response: BMC journals do not provide page numbers in the citation. This is the exact citation provided by the journal.

Comment #43: Ref 8: Page number missing

Authors' response: BMC journals do not provide page numbers in the citation. This is the exact citation provided by the journal.

Comment #44: Ref 9: Please check this

Authors' response: That's the precise citation provided by PLOS One.

Comment #45: Ref 10: Format?

Authors' response: The format fits the journal requirements and we mentioned the full term of the journal.

Comment #46: Ref 12: Page number missing?

Authors' response: BMC journals do not provide page numbers in the citation. This is the exact citation provided by the journal.

Comment #47: Ref 15: Page number missing

Authors' response: BMC journals do not provide page numbers in the citation. This is the exact citation provided by the journal.

Comment #48: Ref 16: Format

Authors' response: As a couple of journals did not have an abbreviation, the full names of all journals were provided. The format is consistent with the journal submission guideline.

Comment #49: Ref 25: Page number missing

Authors' response: BMC journals do not provide page numbers in the citation. This is the exact citation provided by the journal.

Comment #50: Ref 26: Format

Authors' response: As a couple of journals did not have an abbreviation, the full names of all journals were provided. The format is consistent with the journal submission guideline.

Comment #51: Ref 38: Italics needed

Authors' response: According to the submission guideline and example references provided by BMJ Open, no italic has been used in any parts for the 'book' or 'chapter in a book'.

Comment #52: Ref 41: Page number missing

Authors' response: BMC journals do not provide page numbers in the citation. This is the exact citation provided by the journal.

Comment #53: Ref 47: Format

Authors' response: As a couple of journals did not have an abbreviation, the full names of all journals were provided. The format is consistent with the journal submission guideline.

Comment #54: Ref 48: Italics needed

Authors' response: According to the submission guideline and example references provided by BMJ Open, no italic has been used in any parts of the 'book' or 'chapter in a book'.

Comment #55: Table C: Black et al (2022), Study main findings, Please clarify this re study days

Authors' response: The difference between 'employer-paid' and 'personal' study days was clarified as follows.

"... an average of 20.1 employer-paid study days were reported by 92% of nurses and an average of 7.4 supervised days for each nurse (an average cost of £6,45 to the NHS per nurse) during training. Eighty-one percent of nurse prescribers spent an average of 26.3 days of personal time studying for their NMP qualifications."

Comment #56: Table C: Carey et al (2021), Study main findings, Write in full each time please

Authors' response: The full term (i.e. minutes) was provided in Table C as follows:

"Consultation duration: An average of 6.8 minutes higher for physiotherapist prescribers and 3.5 minutes for podiatrist prescribers compared to non-prescribers"

Comment #57: Table C: Neilson et al (2015), Study main findings, Write in full

Authors' response: Although we had provided all the full terms, in the 'Note' below the table, for all the abbreviations used in the table, as requested by the reviewer, the full term for 'EVSI' was provided in Table C.

Many thanks for your valuable comments and suggestions.

Reviewer #2

Comment #1: Thank you for inviting me to review this paper. As the authors, and Noblet et al, report there is a dearth of evidence in this important field.

I have a few comments to make regarding the paper. I note that you registered the protocol in advance, and published it, as well as following the PRISMA-ScR guidelines and I commend you for that.

Authors' response: Thank you for your encouraging remarks. Below, we explain how we have improved our paper based on your comments. The changes are all highlighted in yellow here and in the marked version of the manuscript and supplementary data.

Comment #2: The authors comment correctly that optometrists are non-medical prescribers, but appear to group them under the more general 'allied health professional' (AHP) heading. They are not classed as AHPs (NHS England - <https://www.england.nhs.uk/ahp/role/>), and they aren't regulated by the HCPC, which regulates AHPs, but are in fact regulated by GOC. Please ensure therefore that you refer to optometrists separately throughout the paper. If you could find no studies relating to optometrists, then it would be valuable to state that.

Authors' response: Thanks for your comment. This was revised and we used 'healthcare non-medical professionals' or 'non-medical prescribers' instead and this reference was added when we refer to optometrists.

Comment #3: It was difficult to relate the results section to your very clear research questions that you set out initially. Please rearrange the results section to reflect the order of the questions, as that would aid the reader.

Authors' response: Thanks for your comment. The result section was checked to make sure all research questions are answered in the order they have been presented in the 'Methods' section. Below, we provided more detailed information on how/where each research question is answered in the 'Results' section.

Research question 1) What types of prescribing practices (e.g. SP, IP) have been implemented and evaluated across eligible groups of healthcare professions (e.g. pharmacists, podiatrists, dietitians, etc) in different studies?

Please see page 10, 'General characteristics of included studies' sub-section, lines 49 to 56, as well as Table 3, pages 11 and 12:

'... evaluated the impact of NMP practices by pharmacists (n=4), nurses (n=3),7,50-55 physiotherapists and podiatrists (n=1),15 and another estimating NMP cost-savings in primary and secondary care for a range of health professions.49 Types of prescribing services evaluated in these studies included SP (n=2)49,55 or IP (n=8)7,15,49-54 and community nursing.49'

Research question 2) What measures and tools have been used to evaluate the economic values, safety, effectiveness and other consequences of prescribing by non-medical prescribers in various settings?

Please see page 13, 'Measures of costs' and page 14, 'Measures of outcome' sub-sections.

Research question 3) What are relevant costs, resource use, health, non-health and clinical outcomes associated with services provided by non-medical prescribers in both peer-reviewed and grey literature?

Please see pages 15 to 17, 'Key findings: the costs and consequences of NMP' sub-section.

Comment #4: Please check your use of abbreviations and ensure that they are defined the first time they appear in the text (eg CVD and HbA1C on page 14)

Authors' response: The full terms were provided, and the terms were defined where relevant in the text. For example, see page 14, measures of outcome, line 8: HbA1c test results (mean blood sugar level)

Comment #5: Please check that any references to supplementary data state that, e.g. ref to Table B - Page 9, line 42 and Table C - Page 15, line 9.

Authors' response: This (i.e. Supplementary Data) was added after Tables B and C (for example, see the manuscript, page 9, stage 5, first line).

Comment #6: Page 4, lines 20-27. Your reference numbers jump from 9 to 27. Please check your references as in the numbered system they should follow sequentially when first referenced. Please also ensure that every mention of a study is referenced (e.g. page 15, lines 41-45)

Authors' response: The citations and references were all checked and they are all in order now.

Comment #7: Page 4, Line 44 – 'optometrists' repeated twice

Authors' response: The duplication was removed.

Comment #8: Page 4, Line 46 – comma after '2016' not required

Authors' response: The comma was removed.

Comment #9: Page 6, Line 54, Stage 2. You comment that the scope of NMP ranges and say 'ranging from a restricted formulary to... please clarify what you mean by that.

Authors' response: The sentence was removed to avoid confusion.

Comment #10: Page 7, lines 3-14. Issuing of new prescriptions via a GP system is specifically not mentioned – are these activities not included? (the only GP system related activity appears to be repeat prescribing).

Authors' response: The scope of medicine management and prescribing activities in this scoping review is in line with Carey et al. (2020) (as already explained in the 'search strategy and screening' sub-section, manuscript, page 6. Also, please see Table 1, Carey et al. BMC Health Services Research, 2020, <https://doi.org/10.1186/s12913-020-05918-8>). For this purpose, we are interested in the observations of practice (but not in general practice) as none of the non-medical healthcare professionals (i.e. physiotherapists and podiatrists) assessed in this paper works in the GP system.

Comment #11: Page 9, line 3 – you comment that two reviewers screened the full text articles, and then list 4 reviewers. Please clarify – for example were they split between the authors.

Authors' response: This was clarified as follows (manuscript, stage 3, pages 8 and 9, lines 4 and 5): "The full-text screening of the selected studies was divided between authors and carried out independently by two reviewers"

Comment #12: Page 9, lines 46-52. As it currently reads, it implies that a separate descriptive report was written, in addition to the table. If you mean by 'descriptive report' the narrative in the table, then it would read better as '...tabular form, with a descriptive report...'

Authors' response: This was revised as follows (see the manuscript, stage 5, line 4):

"The findings of selected studies were summarised and presented in tabular forms and descriptively ..."

Comment #13: Page 11-12, Table 3. The legend includes several abbreviations that are not used in the table (but are used in Table C in the supplementary data). Please make sure the legends match the relevant table.

Authors' response: The two tables' legends were checked and revised.

Comment #14: Page 13, line 5. If table C is part of the supplementary data, then this line isn't required as it will be cross linked from the previous reference.

Authors' response: The line referring to Table C was removed.

Comment #15: Page 16, second paragraph. Your comments about PDs not requiring specific training could be interpreted that PDGs are better as the costs are less. Is this correct, or are there limitations to PDGs that mean that NMP is worth the training cost?

Authors' response: This was clarified as follows:

"It is important to note that although PGDs (i.e. patient group directions, please see Table 1 for more information) provide a legal framework for health professionals to supply and administer a specified medicine to a pre-defined group of patients, and there is no mandatory training required prior to their use; there are limitations to their use, indicating that NMP might be worth the training cost."

Comment #16: Page 16 line 37 – comma after PDGs not required

Authors' response: The comma after PDGs was removed.

Comment #17: Page 19, lines 10-16. I'm not clear that your results show that NMP provides value for money (particularly in the nursing group, which comprise the greatest number of NMPs). Please clarify what you mean by this sentence.

Authors' response: The sentence was revised as follows (and 'nurses' was removed to prevent confusion):

"As most cost-effectiveness evidence relates to pharmacists, it is important to evaluate the impact, safety, resource use and economic value of prescribing by non-medical prescribers in other professions ..."

Comment #18: Page 19, line 44-49. The studies you include form before 2015 were not RCTs, and hence were excluded from the paper by Noblet et al. It would be more appropriate to state that you have included '...findings from studies that were published...' and '... as not RCT studies, were excluded from the previous review...' or something similar.

Authors' response: This line was removed.

Thank you for all your valuable comments and suggestions.

VERSION 2 – REVIEW

REVIEWER	McIntosh, Trudi Robert Gordon University, Pharmacy and Life Sciences
REVIEW RETURNED	17-Nov-2022

GENERAL COMMENTS	Many thanks for your revisions addressing my comments so thoroughly, and those of the other reviewer. I think the paper reads very well now, is clear for non-specialist readers and makes a valuable contribution to the literature. I'm looking forward very much to seeing it 'in print'.
--

REVIEWER	Graham-Clarke, Emma
-----------------	---------------------

	Sandwell and West Birmingham Hospitals NHS Trust, Anaesthetic Department
REVIEW RETURNED	16-Nov-2022

GENERAL COMMENTS	Thank you for addressing my previous comments. I have one further comment to make for your consideration. I can see that you have ordered the results section according to your research questions. However, your sub-headings in the results section use different terminology to the questions and this makes it more difficult to follow. For example question 3 uses the phrase 'clinical outcomes', but outcomes appear in the results section relating to question 2 as 'measures of outcome'. In comparison, 'consequences' is used in question 2, but appears in the sub-heading for the results relating to question 3. Your results would be easier to follow if your subheadings in the results section reflected your questions more closely.
---

VERSION 2 – AUTHOR RESPONSE

Reviewer #1

Comment #1: Many thanks for your revisions addressing my comments so thoroughly, and those of the other reviewer. I think the paper reads very well now, is clear for non-specialist readers and makes a valuable contribution to the literature. I'm looking forward very much to seeing it 'in print'.

Authors' response: Thank you for your encouraging remarks.

Reviewer #2

Comment #1: Thank you for addressing my previous comments. I have one further comment to make for your consideration. I can see that you have ordered the results section according to your research questions. However, your sub-headings in the results section use different terminology to the questions and this makes it more difficult to follow. For example question 3 uses the phrase 'clinical outcomes', but outcomes appear in the results section relating to question 2 as 'measures of outcome'. In comparison, 'consequences' is used in question 2, but appears in the sub-heading for the results relating to question 3. Your results would be easier to follow if your subheadings in the results section reflected your questions more closely.

Authors' response: Thank you for your comments. Below, we explained how we made changes based on your comment. The changes are all highlighted in yellow here and in the marked version of the manuscript.

In the method section, Stage 1: Identifying the research questions, we added the term 'consequences', as follows:

"3) What are relevant costs, resource use, and consequences (e.g. health, non-health and clinical outcomes) associated with services provided by non-medical prescribers in both peer-reviewed and grey literature?"

Other changes include adding the terms 'of NMP' to 'measures of costs' in the result section as appears below:

"Measures of costs of NMP"

Also, we changed 'outcomes' to 'consequences' in the subtitle you mentioned in the result section to be consistent with the terminologies used in the research questions, as follows:

"Measures of consequences of NMP"

Thank you for all your valuable comments and suggestions.